# *Lacticaseibacillus casei* Strain Shirota Modulates Macrophage-Intestinal Epithelial Cell Co-Culture Barrier Integrity, Bacterial Sensing and Inflammatory Cytokines

**DOI:** 10.3390/microorganisms10102087

**Published:** 2022-10-21

**Authors:** Andrew Foey, Neama Habil, Alex Strachan, Jane Beal

**Affiliations:** 1School of Biomedical Sciences, University of Plymouth, Drake Circus, Plymouth PL4 8AA, UK; 2Peninsula Dental School, University of Plymouth, Drake Circus, Plymouth PL4 8AA, UK; 3School of Biological & Marine Sciences, University of Plymouth, Drake Circus, Plymouth PL4 8AA, UK; 4School of Life, Health & Chemical Sciences, Faculty of Science, Technology, Engineering & Mathematics, The Open University, Milton Keynes MK7 6AA, UK

**Keywords:** probiotics, macrophages, epithelial cells, cytokines, inflammation

## Abstract

Probiotic bacteria modulate macrophage immune inflammatory responses, with functional cytokine responses determined by macrophage subset polarisation, stimulation and probiotic strain. Mucosal macrophages exhibit subset functional heterogeneity but are organised in a 3-dimensional tissue, over-laid by barrier epithelial cells. This study aimed to investigate the effects of the probiotic *Lacticaseibacillus casei* strain Shirota (LcS) on macrophage-epithelial cell cytokine responses, pattern recognition receptor (PRR) expression and LPS responses and the impacts on barrier integrity. THP-1-derived M1 and M2 subset macrophages were co-cultured in a transwell system with differentiated Caco-2 epithelial cells in the presence or absence of enteropathogenic LPS. Both Caco-2 cells in monoculture and macrophage co-culture were assayed for cytokines, PRR expression and barrier integrity (TEER and ZO-1) by RT-PCR, ELISA, IHC and electrical resistance. Caco-2 monocultures expressed distinct cytokine profiles (IL-6, IL-8, TNFα, endogenous IL-10), PRRs and barrier integrity, determined by inflammatory context (TNFα or IL-1β). In co-culture, LcS rescued ZO-1 and TEER in M2/Caco-2, but not M1/Caco-2. LcS suppressed TLR2, TLR4, MD2 expression in both co-cultures and differentially regulated NOD2, TLR9, Tollip and cytokine secretion. In conclusion, LcS selectively modulates epithelial barrier integrity, pathogen sensing and inflammatory cytokine profile; determined by macrophage subset and activation status.

## 1. Introduction

Mucosal macrophages are integral to the immune-fate decisions, as inducing a hypo-responsive tolerance or activating an appropriate anti-pathogen response; effectively acting as an immune on–off switch that determines mucosal responses to the local environment of the gastrointestinal tract [1]. These innate immune cells effectively discriminate between safe, non-self, such as antigens derived from digested food products and commensal or probiotic microbes, and unsafe, non-self, factors such as pathogenic microbes and harmful antigens. These immune fate decisions made by mucosal macrophages are determined by local environmental factors and the effect they exert on macrophage differentiation, plasticity and activation [2]. The local environment encountered by these macrophages can therefore prime either towards a tolerogenic, anti-inflammatory response or an immune-activatory pro-inflammatory response. This dichotomy in immune response is reflected by the plasticity and polarisation of these mucosal macrophages to distinct functional subsets. In general, homeostatic mucosal macrophages resemble the M2 subset and are hypo-responsive and exhibit a phenotype characterised by high phagocytic activity and secretion of anti-inflammatory cytokines such as IL-10 [3,4,5]. Macrophages encountering pathogenic microbes are functionally similar to the M1 subset that is predominated by the secretion of pro-inflammatory cytokines capable of perpetuating innate inflammatory responses and activating distinct T helper subsets critical for anti-microbial defences, which if uncontrolled, results in host tissue-destructive pathology.

The addition of an exogenous source of beneficial microbes, such as probiotic Lactic Acid Bacteria (LAB), has been demonstrated to directly modulate macrophage functionality in a strain-specific and subset-dependent manner [reviewed in [6]. Indeed, *Lacticaseibacillus casei* strain Shirota (LcS) exhibits a modulatory effect on both pro-inflammatory and anti-inflammatory cytokine secretion, as well as the anti-microbial peptide (AMP), human beta-defensin 2 (hBD-2). Heat-killed LcS was described to augment LPS-induced TNFα secretion by CD14^hi^ M1 macrophages, whereas it suppressed LPS-induced TNFα and IL-6 in CD14^lo^ M2 macrophages. These modulatory effects on pro-inflammatory cytokine secretion were dependent on macrophage subset, CD14 (the TLR2 and TLR4 co-stimulatory molecule) expression levels and probiotic strain being investigated [7,8].

With respect to mucosal barrier cells, such as intestinal epithelial cells (IECs), again the immune response is dependent on the local environment. HK-LcS induces Caco-2 colonic epithelial cell hBD-2 expression and secretion; however, this probiotic effect on hBD-2 is determined by inflammatory context [9], where HK-LcS augmented both IL-1β- and TNFα-induced hBD-2 and differentially rescued the IL-10 suppression of this AMP. In addition, the expression and secretion of both pro-inflammatory and anti-inflammatory cytokines by epithelial cells is affected by this probiotic strain. LcS differentially modulates epithelial cell TNFα (pro-inflammatory) and IL-10 (anti-inflammatory) dependent on induction stimulus (TNFα or IL-1β).

Discrimination between pathogenic and non-pathogenic bacteria in the gastrointestinal tract requires the crosstalk between IECs and underlying immune cells [10]. Much attention has been invested in developing physiologically relevant co-culture models of the gut mucosa, using epithelial cells and macrophages [11,12,13]. In addition, the degree of epithelial cell (Caco-2) polarisation and differentiation determines how the mucosal barrier responds to and determines entry of pathogenic microbes such as *Listeria monocytogenes* [14]. In general, Caco-2 cells are representative of colonic IECs whereas as a model of the small intestine, Caco-2 cells require differentiation and can also be used in co-culture with HT29-5M21 mucin-secreting intestinal cells [15]. Original epithelial cell barrier studies utilised co-culture of Caco-2 cells with PBMCs, investigating the effect of *Escherichia coli* (*E. coli*) bacteria on PBMC cytokine production. It was found that epithelial cells suppressed basolateral stimulation of PBMC secretion of TNFα and IL-12, whereas they exerted no effect on induction of IL-10, IL-1β, IL-6, IFNγ, IL-8 and TGFβ_1_ [16]. Thus, possibly controlling inflammatory and cell-mediated immune responses. This suppression of pro-inflammatory cytokines may also reflect that epithelial cell-immune cell interactions may mediate the ability of these mucosal cells to detect pathogens by profoundly affecting both PRR expression and signalling. Indeed, human IECs constitutively express TLR3, TLR5, TLR9, NOD1, NOD2 and low levels of TLR2 and TLR4, where TLR4 is increased in inflammatory bowel disease (IBD) IECSs and intestinal macrophages (reviewed in [17]). IECs are generally hyporesponsive to TLR2-mediated signalling induced by PAMPs such as peptidoglycan (PGN) and lipoteichoic acid (LTA), whereas maintaining a responsiveness to TLR4 ligands such as LPS [18]. This hyporesponsiveness was maintained by TLR2 absence at the cell membrane and a corresponding high-level expression of the negative regulator, Toll inhibitory protein (Tollip). It is conceivable that mucosal cell interactions, such as those between IECs of the barrier layer and mucosal macrophages, will play an important role in determining mucosal bacterial sensing and immune responsiveness to such microbes, which is both determined by and determines the barrier integrity of the intestinal mucosa.

These studies have highlighted that barrier defences of mucosal epithelial cells and immune defences of macrophage cells are highly variable, indicated by a range of markers, which include the expression, secretion and activity of pro- and anti-inflammatory cytokines, expression and functionality of AMPs and pathogen recognition by pattern recognition receptors. All of these factors are likely to come into play to determine immune on/off fate, and how they respond to beneficial microbes such as exogenously added probiotics. As such, the 3D mucosal tissue organotypic context in which macrophages recognise beneficial microbes is thus imperative to these immune fate decisions. The aim of this study was to investigate the immunomodulatory effect of the probiotic, LcS, on epithelial cell—macrophage co-cultures with respect to mucosal barrier integrity, bacterial sensing and inflammatory cytokine production. Such studies will further the understanding of immunomodulatory effects of probiotics in tissue-mimetic models, as opposed to cell monoculture models which may prove to be insufficient to comprehensively understand how probiotics modulate mucosal responses of the gut.

## 2. Materials and Methods

### 2.1. Bacterial Culture and Preparation of Heat Killed Extracts

*Lacticaseibacillus casei* strain Shirota (LcS) was obtained from a commercial probiotic product (Yakult Ltd.). Probiotic bacteria were cultured in deMan Rogosa Sharp (MRS) broth at 37 °C for 18 h until the commencement of stationary phase. The preparation of heat killed (HK) bacterial samples were made according to the method described in [19]. In brief, bacterial cells were centrifuged and washed twice in phosphate-buffered saline and viable counts adjusted to a density of 1 × 10^9^ cfumL^−1^. Probiotic bacteria were heat killed (HK) for 2 h at 65 °C. To confirm death of bacteria, all *Lactobacillus* spp. Samples were plated on MRS agar and incubated for a minimum of 18 hrs. All HK bacterial extracts were stained by Gram stain to check bacterial integrity after heating.

### 2.2. Cell Culture & Co-Culture

Caco-2 (Human colon adenocarcinoma) epithelial cell line was kindly provided by Dr. Maria O’Connell, HNR, Cambridge, UK. Caco-2 cells were maintained in D10 medium, Dulbecco’s Modified Eagles’ Medium (DMEM) supplemented with 10% *v*/*v* foetal calf serum, 2 mM L-glutamine and 100 UmL^−1^ penicillin, 100 µgmL^−1^ streptomycin (Lonza, Wokingham, UK). Adherent cells were removed from tissue culture plastic by washing three times in sterile DPBS, followed by treatment with 0.25% *v*/*v* trypsin-EDTA. Harvested cells were counted and density adjusted to 5 × 10^5^ cellsmL^−1^ and plated/cultured in D10 medium in a humidified atmosphere at 37 °C, 5% CO_2_ for 21 days to allow for full cell differentiation [20] and an intact barrier, as defined by measurement of Trans-epithelial electrical resistance (TEER).

THP-1 pro-monocytic cells were maintained in R10 medium (RPMI-1640 medium supplemented with 10% *v*/*v* foetal calf serum, 2 mM L-glutamine and 100 UmL^−1^ penicillin 100 mgmL^−1^ streptomycin. Cells were passaged every 3–4 days and routinely used for experimentation between passages 15–35. M1-like (pro-inflammatory subset) and M2-like (anti-inflammatory/regulatory) THP-1-derived macrophages were generated by differentiating THP-1 cells, seeded to a density of 1 × 10^6^ cellsmL^−1^, in 25 ng/mL PMA (Sigma-Aldrich, Poole, UK) and 10 nM 1,25-(OH)_2_ Vitamin D_3_ (Sigma-Aldrich, Poole, UK) for 3–4 days and 7–8 days, respectively, according to the protocol detailed in [7,8,9,21,22,23,24,25,26,27,28,29,30,31,32], and resulting in the M1- and M2-like phenotypes (M1-like: iNOS^hi^ Arg^−^ CD206^−^ Phagocytosis^lo^ TLR4^hi^ TNFa^hi^ IL-6^hi^ IL-8^hi^ IL-10^lo^; M2-like: iNOS^lo/neg^ Arg^+^ CD206^+^ Phagocytosis^hi^ TLR4^lo^ TNFα^lo^ IL-6^lo^ IL-8^lo^ IL-10^hi^) and further detailed in Appendix A.

Co-cultures were set up to mimic both homeostatic and inflammatory environments of the intestinal mucosal lining. Caco-2 cells were seeded at a density of 5 × 10^5^ cells/well in transwell cell culture inserts (pore size 1.0 μm, Becton Dickinson, NJ, USA) in D10 medium for 21 days at 37 °C, 5% CO_2_ for full differentiation to intestinal epithelial cells. For the final day prior to experimental start, D10 was removed from the Caco-2 cells and replaced by R10 medium. Previously prepared THP-1-derived M1 and M2 macrophages were introduced in the basolateral compartment of the tissue culture well to a density of 1 × 10^6^ cellsmL^−1^ in R10 medium. Epithelial cells were apically applied with heat-killed LcS probiotic bacteria at a density of 1 × 10^8^ cfumL^−1^ for 18 hrs. To mimic mucosal disruption in an inflammatory environment, K12-LPS (100 ng/mL) was applied below the transwell in the basolateral compartment, containing either M1 or M2 subset macrophages, for a further defined incubation period.

### 2.3. Modulatory Effect of Heat-Killed LcS Probiotic Bacteria on Epithelial Cell Cytokine Production

Differentiated epithelial cells were grown to confluence (intact mucosal barrier) and pre-treated in the presence or absence (R10 medium alone control) of heat-killed LcS probiotic bacteria at a pre-defined optimal density of 1 × 10^8^ cfumL^−1^ density equivalents for 18 h prior to stimulation. To negate any effects of lactic acid produced by live LcS on cell viability, hence indirectly modulating cytokine responses, heat-killed bacteria were adopted for subsequent experimentation. Caco-2 cells were stimulated with the pro-inflammatory cytokines prevalent in inflammatory bowel disease, 10 ng/mL TNFα or 10 ng/mL IL-1β for a further pre-determined time point of 18 h, optimal for pro-inflammatory cytokine stimulation and heat-killed probiotic bacteria (LcS) to induce cytokine mRNA expression and protein secretion from Caco-2 cells. TNFα-induction of TNFα protein is problematic when measuring secretion of TNFα protein; to circumvent this issue with this stimulation protocol, Caco-2 cells were stimulated using a pulse-chase protocol (for both stimuli). Cells were stimulated by 10 ngmL^−1^ exogenously added TNFα (or IL-1β) for 6 hrs, after which time the TNFα/cytokine stimulus was washed off the cells, replaced by fresh R10 medium and incubated (chased) up to 18 hrs. At the end of experiment, both cells (for cell lysate) and conditioned medium supernatant were harvested for assay of secreted cytokine proteins and mRNA expression. Cell viability of differentiated Caco-2 epithelial cells was routinely monitored by MTT assay and found to be unaffected by the concentrations of reagents used and the incubation times adopted.

### 2.4. Detection of Endogenous IL-10 Activity and Its Effect on Modulation of Epithelial Cell Bacterial Sensing Molecules

In order to determine whether Caco-2 IECs exhibit an endogenous, cell-membrane associated IL-10 activity and if bacterial sensing molecules were associated with endogenous IL-10 activity, the ability of IL-10 to act as an anti-inflammatory cytokine and suppress the secretion of the pro-inflammatory cytokine, TNFα, has been harnessed. If endogenous IL-10 activity exists, suppression of this activity should augment induction of pro-inflammatory TNFα secretion, as such neutralising antibodies were employed. Caco-2 cells were either unstimulated or stimulated by either 10 ng/mL TNFα or IL-1β in the presence or absence of the neutralising antibody: anti-human IL-10 (clone JES3-9D7, Biolegend, San Diego, CA, USA; controlled for by an irrelevant isotype-matched antibody) at a concentration of 10 µgmL^−1^ (previously determined to neutralise 10 ngmL^−1^ of IL-10) [33]. Epithelial cells were incubated for a further 18 hrs, after which time the cells were washed and collected for mRNA extraction and quantification of both TNFα cytokine and bacterial sensing molecule gene expression (TLR2, TLR4, MD-2, CD14, NOD2, TLR9 and Tollip), and culture supernatants were collected to detect secreted TNFα cytokine protein by ELISA.

### 2.5. Measurement of Trans-Epithelial Electrical Resistance (TEER)

Semi-permeable membrane transwell inserts containing confluent differentiated Caco-2 IEC membrane layer in the transwell (apical) compartment were carefully removed from co-culture at the end of the experiment. After removal of apical supernatant, the Caco-2 cell mucosal layer was washed twice by DPBS. A volume of 0.5 mL DPBS was added in the transwell and 1.0 mL added to the basolateral compartment outside the transwell insert in the EVOM Epithelial Voltameter (Pharma, UK). Trans-epithelial electrical resistance (TEER) was recorded by the voltameter across the semi-permeable/cellular barrier in the insert and adjusted for the transwell surface area of 0.33 cm^2^ to present as Ω cm^2^. All readings of TEER were repeated across triplicate sample transwells.

### 2.6. Immunohistochemical (IHC) Localisation of Epithelial Cell Zona Occludin-1 (ZO-1)

Caco-2 transwell monolayers were apically washed using DPBS. Cells were then fixed with 3% *w*/*v* paraformaldehyde in CS buffer (0.1 M NaOH, 0.1 M HEPES, 1 mM EGTA, pH 6.8) for 20 min, followed by three, 5 min washes using CS buffer and permeabilised with 0.1% *v*/*v* Triton X-100. Epithelial cells were then blocked by 1% *w*/*v* BSA in DPBS for 1 hr and followed by overnight incubation with primary antibody (2.5 μg/mL rabbit polyclonal anti-ZO-1, Invitrogen, UK) in 1% *w*/*v* BSA/DPBS at 4 °C. Cell monolayer was then washed three times with CS buffer followed by 1 h incubation in secondary antibody (0.5 μgmL^−1^ Alexa fluor 488-conjugated anti-rabbit IgG, Invitrogen, UK) in 1% *w*/*v* BSA/DPBS. All monolayers were washed three times in CS buffer then mounted in DPX on glass slides and visualised by a Nikon Eclipse 80i epifluorescence microscope with QiMc camera using NIS-Elements Software (Nikon DS-BR 3.0, Nikon, Melville, NY, USA).

### 2.7. Detection of Membrane TLR2 and TLR4 Protein by Flow Cytometry

Caco-2 IECs were harvested and resuspended to a density of 2 × 10^6^ cellsml^−1^; 100 μL aliquots were washed twice in sterile DPBS and incubated with 2% *w*/*v* BSA in Ca^2+^/Mg^2+^ free PBS for 30 min on ice to enable blocking of non-specific binding of antibodies. Cells were washed in 1% *w*/*v* BSA/PBS and incubated with the appropriate PE fluorochrome-conjugated antibody (1/200 dilution anti-TLR2 Clone TL2.1, anti-TLR4 Clone HTA125, and appropriate isotype-matched control antibodies)(eBiosciences, UK) for 30 min at 4 °C in the dark. Cells were washed twice in 1% *w*/*v* BSA/PBS to remove unbound antibody and resuspended in 500 μL 1% *w*/*v* BSA/PBS and kept on ice in the dark, until ready for data acquisition and analysis. TLR2 and TLR4 staining was detected by analysing the cell samples in a FACS Calibre Flow Cytometer (Becton Dickinson, San Jose, CA, USA) and analysed by BD FACS Diva Software v6.0. Positive staining for these membrane receptors is expressed as Net mean fluorescence intensity (Net MFI) obtained by subtracting isotype control sample MFI from positive test MFI for at least 3 replicate samples live gated by FSC/SSC and positivity determined by setting 5% confidence interval gating on live cells.

### 2.8. Real Time qPCR Analysis

The expression of TNFα, IL-6, IL-8, IL-10, TLR2, TLR4, MD-2, CD14, NOD2, TLR9, Tollip, ZO-1 and GAPDH mRNA were assessed by quantitative real-time PCR. Following each treatment, the cells were washed with ice-cold PBS and the total RNA was extracted using Sigma genelute RNA isolation kit (Sigma-Aldrich, Poole, Dorset, UK) according to the manufacturer’s instruction. The total RNA concentration was determined using NanoVue^TM^ spectrophotometer (GE Healthcare, Freiberg, Germany). RNA purity was assessed by examining the absorbance ratio at 260 and 280 nm, while the integrity was verified by electrophoresis on 1% denaturing agarose gel. A microgram of total RNA was reverse transcribed using M-MLV Reverse Transcriptase reaction Kit (Sigma-Aldrich) as suggested by the supplier. Sequence-specific primers (Table 1) were designed using Primer Express Software (Applied Biosystems, Paisley, UK) and synthesized by Eurofin MWG/ Operon (Ebersberg, Germany). The quantitative real-time PCR was performed using StepOnePlus thermal cycler and Power SYBR Green^®^ kit (Applied Biosysterms) using 10 pmol of the forward and reverse primers for each target. The amplification of target was carried out under the following conditions: pre-heating for 95 °C for 10 min, followed by 40 cycles at 95 °C for 30 s, 60 °C for 1 min and 72 °C for 1 min. The real time quantitative PCR data was analysed following the 2^−ΔΔCt^ method as described by in [34] using GAPDH as an endogenous control and resting cells as a reference sample. Thus, the relative quantity of the target transcript is described as fold increase (RQ, relative quantification) relative to the reference sample and GAPDH.

### 2.9. Quantification of Cytokine Secretion

Cytokine secretion of IL-6, IL-8, IL-10 and TNFα, into cell culture supernatants by treated Caco-2 IECs and Caco-2/macrophage-subset co-cultures were quantified by sandwich ELISA, using commercially available paired capture and detection antibodies for TNFα, IL-6, IL-8 and IL-10 (BD-Pharmingen, Oxford, UK). Protocols were followed according to manufacturer’s instructions and compared to standard curves of recombinant human cytokines (using recognised international cytokine standards available from NIBSC, Potter’s Bar, UK) between the range of 7 to 5000 pgmL^−1^. Colorimetric development was determined spectrophotometrically by an OPTIMax tuneable microplate reader at 450 nm and analysed by Softmax Pro version 2.4.1 software (Molecular Devices Corp., Sunnyvale, CA, USA).

### 2.10. Statistical Analysis

All data are expressed as the mean ± standard error of triplicate values of a representative experiment of at least three independent experiments. Statistical analysis was performed using one-way analysis of variance (ANOVA). Statgraphic Software version 5.1 was used to analyse the data. Statistical significance is indicated by a *p* value of less than 0.05, where * *p*< 0.05, ** *p*< 0.01,*** *p* < 0.005 versus stimulation control.

## 3. Results

### 3.1. Modulation of IEC Cytokines by Heat-Killed LcS Is Dependent on Inflammatory Environment

Pro-inflammatory cytokines are known to induce cytokine expression and secretion by Caco-2 IECs. The role of LcS in modulating the responses of these cells in defined inflammatory environments is relatively superficial, yet important to understand monoculture responses when building up to co-culture and 3D organotypic culture beyond. TNFα clearly induced the expression and secretion of TNFα, IL-8 and IL-10, whereas IL-1β-stimulated Caco-2 cells produced significant levels of all IL-6, IL-8, TNFα and IL-10. HK-LcS significantly augmented TNFα-induced IL-8, TNFα and IL-10 mRNA by 53- (*p* < 0.01), 3.6- (*p* < 0.001) and 20-fold (*p* < 0.01), respectively, but failed to significantly modulate the protein secretion of the aforementioned cytokines. In the case of a pro-inflammatory environment induced by IL-1β, however, LcS augmented IL-6, IL-8 and TNFα mRNA by 33% (*p* < 0.05), 170% (*p* < 0.01) and 400% (*p* < 0.001), respectively, whereas IL-10 mRNA was suppressed by 80% (*p* < 0.05). Additionally, LcS augmented IL-6 and IL-8 protein secretion by 114% (*p* < 0.01) and 26% (*p* < 0.05), respectively, and IL-10 secretion was suppressed by 56%, *p* < 0.001 (see Table 2).

It is important to note that, although HK-LcS clearly modulated IL-10 mRNA, these trends were not represented when measuring secreted IL-10 protein. Endogenous, membrane-bound IL-10 may account for this discrepancy in IL-10 mRNA versus secreted protein levels and has been previously described in macrophages and epithelial cells [9,35,36]. As a consequence, endogenous IL-10 was measured, based on the ability of a neutralising anti-IL-10 antibody to suppress the anti-inflammatory effects of IL-10 on the secretion of the pro-inflammatory cytokine, TNFα [37]. Data presented in Table 3 demonstrates that unstimulated Caco-2 IECs constitutively express a basal endogenous IL-10 activity, as judged by the increase in TNFα mRNA and protein secretion upon treatment with neutralising anti-IL-10 (*p* < 0.001). Importantly, in a pro-inflammatory environment defined by TNFα, endogenous IL-10 activity was further elevated (difference between untreated 80 pg/mL and anti-IL-10 treated 710 pg/mL, *p* < 0.001), whereas in the case of IL-1β-stimulated Caco-2 cells, neutralisation of IL-10 resulted in a decrease in TNFα secretion from 330 pg/mL to 250 pg/mL (*p* < 0.01).

To further define the importance of epithelial cell endogenous IL-10 activity, the neutralising anti-IL-10 antibody was investigated for its effects on bacterial sensing molecules that detect bacterial PAMPs and regulate their responses upon detection (see Table 4). Neutralisation of IL-10 bioactivity in unstimulated Caco-2 cells demonstrated a homeostatic endogenous IL-10 bioactivity, which negatively regulated TLR2, TLR4, CD14, MD2, NOD2 and TLR9 as demonstrated by the augmentation of mRNA expression (ration of mRNA anti-IL-10/Control) by 100-, 50-, 70-, 3-, 50-, 90-fold, respectively. Of particular note, however, is the positive regulation of the negative regulatory molecule, Tollip (ratio 0.001:1). In an inflammatory environment, induced by TNFα, the anti-IL-10/Control ratios were all positive, indicative of negative regulation by IL-10, but were reduced to 4, 24, 10, 4 for TLR4, CD14, NOD2 and TLR9, respectively. TLR2 ratio (100:1) was unchanged, whereas MD2 and Tollip were increased to 4 and 2, respectively. In the presence of IL-1β, anti-IL-10 treatment upregulated Caco-2 mRNA expression of TLR2 and Tollip to 180 and 10, respectively, whereas it suppressed the ratios of TLR4, CD14, MD2, NOD2 and TLR9 to 5, 1, 0.5, 7 and 10, respectively, when compared to unstimulated epithelial cells.

### 3.2. Macrophage Subset and Inflammatory Cytokines Determine IEC Barrier Integrity

Co-culture of macrophage subsets (basolateral plate well) with Caco-2 IECs (apical, transwell) induces a cytokine profile determined by macrophage subset. In the absence of exogenous stimuli, co-culture of homeostatic M2-like macrophages with Caco-2 cells results in the induction of IL-8 (700 pg/mL), whereas TNFα, IL-6 and IL-10 secretion was barely detectable. On the other hand, M1 pro-inflammatory macrophages co-cultured with Caco-2 cells resulted in appreciable levels of TNFα, IL-6 and IL-8 secretion (1000–4000 pg/mL), with IL-10 not detected (Figure 1a). This M1 co-culture cytokine profile is likely to have significant effects on epithelial barrier integrity; where Caco-2 barrier integrity, measured by transepithelial electrical resistance (TEER), is partially reduced by co-culture with M2 macrophage subset (from 920 to 710 Ωcm^2^) whereas integrity is greatly reduced upon co-culture with the pro-inflammatory M1 subset (reduced from 920 Ωcm^2^ to 240 Ωcm^2^)(Figure 1b). This destructive effect on barrier integrity TEER is reproduced when treating Caco-2 cells with the pro-inflammatory cytokines, TNFα IL-1β and IL-8 (reduction in TEER by 90% (*p* < 0.001), 52% (*p* < 0.001) and 75% (*p* < 0.001), respectively), whereas the anti-inflammatory cytokine, IL-10, augmented TEER by 14%, *p* < 0.05 (Figure 1c). This result was further reinforced when measuring ZO-1 mRNA gene expression; ZO-1 (a tight junction protein, which maintains barrier integrity) is significantly suppressed by TNFα (*p* < 0.01), IL-1β (*p* < 0.05) and IL-8 (*p* < 0.001) and enhanced by IL-10 (*p* < 0.05) (Figure 1d). With regards ZO-1 protein, immunohistochemical localisation demonstrates clear localisation of ZO-1 at the epithelial cell boundaries joining cells coming together at ruffled edges (Control, Figure 1e). This boundary staining of ZO-1 was appreciably reduced in Caco-2 cells co-cultured with M1 macrophages (+M1 Mφs, Figure 1f) and in the presence of IL-1β (+M1 Mφs + IL-1β, Figure 1h), with cell boundary ruffling being rounded and staining intensity reduced. In the case of co-culture with M1 macrophages in the presence of LPS, ZO-1 staining was extremely reduced in both intensity and loss of ruffling of the cell boundary interlinkages (Figure 1g). No such reduction in ZO-1 staining intensity and boundary ruffling was observed for co-culture with M2 subset cells (Figure 1i). Addition of LPS (Figure 1j) or IL-1β (Figure 1k) did not appreciably reduce intensity of ZO-1 staining but did have an effect of rounding epithelial cells resulting in loss of boundary ruffling between epithelial cells.

### 3.3. LcS Differentially Rescues Epithelial Barrier Integrity in Macrophage Subset-IEC Co-Cultures

With the establishment of the effects of M1 and M2 co-culture with IECs on cytokine environment and the consequent effects on epithelial barrier integrity, it was important to investigate whether the heat-killed probiotic strain LcS could modulate these macrophage-IEC co-culture responses. LcS exhibits a differential response in modulating barrier integrity of Caco-2 epithelial cells, when co-cultured with either M1 or M2 subset macrophages. LcS failed to rescue TEER in M1-Caco-2 co-culture, whereas augmented TEER from control levels of 650 Ω cm^2^ to 810 Ω cm^2^ in M2-Caco-2 co-culture, compared to Caco-2 alone control TEER of 730 Ω cm^2^ (Figure 2a). In the absence of pathological stimulation, LcS augmented ZO-1 mRNA expression to RQ values of 4 and 33 for M1-Caco-2 and M2-Caco-2 co-cultures, respectively, with M2 co-culture expressing approximately 8-fold more ZO-1 (Figure 2c). This differential rescue response on barrier integrity was reproduced in the presence of LPS stimulation, where LcS augmented TEER by 18% (*p* < 0.05) in M2-Caco-2 co-culture (Figure 2b) and augmented ZO-1 expression in M2-Caco-2 co-culture, *p* < 0.05 (Figure 2d). Notably, LPS dramatically reduced Caco-2 ZO-1 expression in M1 co-cultures to 1/40,000 control levels, which was rescued by LcS (*p* < 0.001), but to a level approximately 1/1000 of control expression (Figure 2d), paralleled by a non-significant rescue of LPS-stimulated M1/Caco-2 barrier integrity TEER (Figure 2b).

### 3.4. LcS Differentially Modulates IEC LPS-Sensing Molecules in Macrophage Subset Co-Culture

Inflammatory environments are known to modulate bacterial sensing and inflammatory responses of IECs. Both macrophages and epithelial cells can respond to LPS, derived from Gram-negative bacteria. Such pathogens are associated with chronic inflammatory bowel disease and barrier disruption; it was imperative to investigate the effect of LcS on LPS-induced PRRs and sensing molecules in these macrophage subset co-cultures. LPS clearly induced TLR4 expression in both co-culture models which was highest in M1 co-culture than the M2 co-culture (RQ: 90 vs. 20 for M1 and M2, respectively) (Figure 3a). LcS suppressed LPS-induced response by 20% and 50%, respectively. This trend of LcS-induced suppression of LPS-induced TLR4 was reproduced when measuring TLR4 protein by surface measurement by flow cytometry (Figure 3b), where M1/Caco-2 TLR4 was suppressed by 20% (*p* < 0.05) and M2/Caco-2 by 85% (*p* < 0.001). Interestingly, the TLR4 co-receptor, CD14 was differentially modulated, where LcS failed to suppress M1/Caco-2 CD14 expression and suppressed M2/Caco-2 CD14 by 30% (*p* < 0.05) (Figure 3c). As with TLR4, LPS-induced MD-2 expression was suppressed by LcS by 30% (*p* < 0.05) and 70% (*p* < 0.05) in M1/Caco-2 and M2/Caco-2 co-cultures, respectively (Figure 3d). With respect to TLR2 mRNA, LcS augmented M1/Caco-2 expression by 57% (*p* < 0.05) whereas suppressed that in M2/Caco-2 by 86% (*p* < 0.001) (Figure 3e). This trend was not replicated in TLR2 protein, where LcS suppressed membrane protein in both co-cultures by 63% and 40%, respectively (*p* < 0.05)(Figure 3f). A similar trend to TLR2 mRNA was observed for NOD2 mRNA; LcS augmented M1/Caco-2 expression by 440% (*p* < 0.001) and suppressed M2/Caco-2 by 90% (*p* < 0.001) (Figure 3g). This differed with TLR9 mRNA gene expression, where LcS suppressed that in M1/Caco-2 by 93% (*p* < 0.001) and augmented in M2/Caco-2 by 106% (*p* < 0.001)(Figure 3h). Finally, LcS significantly rescued M1/Caco-2 expression of the negative regulator, Tollip by 33% (*p* < 0.05), but to a level barely expressed in comparison to the unstimulated M1/Caco-2 control level. In addition, LcS was unable to rescue the LPS-induced down-regulation of Tollip expression observed with M2/Caco-2 co-culture (ns) (Figure 3i).

### 3.5. LcS Modulation of Co-Culture Pro- and Anti-Inflammatory Cytokines Is Dependent on Macrophage Subset and Stimulation Status

Consequent to LcS modulation of bacterial sensing pattern recognition receptors, it was important to investigate the downstream effects on cytokine secretion by these macrophage-IEC co-cultures in both an environment stimulated and unstimulated by enteropathogenic LPS. Such stimulation would occur upon barrier disruption associated with chronic IBD. In the absence of LPS stimulation, LcS differentially modulated cytokine production, where M1-Caco-2 co-culture IL-6 (*p* < 0.05), IL-8 (*p* < 0.01) and IL-10 (*p* < 0.05) were suppressed and TNFα (ns) was unchanged (Figure 4a–d). In M2-Caco-2 co-culture, LcS also suppressed IL-8 by 50% (*p* < 0.01) but in contrast, augmented TNFα (*p* < 0.05), IL-6 (*p* < 0.001) and IL-10 secretion (*p* < 0.01) (Figure 4a–d). In the presence of LPS stimulation, LcS partially augmented M1-Caco-2 co-culture secretion of TNFα (*p* < 0.05) (Figure 4e) and clearly augmented IL-6 (*p* < 0.01) whereas IL-8 (*p* < 0.001) and IL-10 (*p* < 0.05) were suppressed (Figure 4f–h). In the case of M2-Caco-2 co-culture, LcS suppressed LPS-induced TNFα and IL-8 (*p* < 0.05) (Figure 4e,g), augmented IL-6 (*p* < 0.05) (Figure 4f) and failed to modulate IL-10 secretion (ns) (Figure 4h).

## 4. Discussion and Conclusions

This investigation demonstrated several aspects of modulation of barrier functionality in the context of IECs alone or in co-culture with defined macrophage subsets. In the case of epithelial cell monoculture, (1) LcS differentially modulates pro- and anti-inflammatory cytokines dependent on inflammatory cytokine environment induced by either TNFα or IL-1β, (2) TNFα and IL-1β differentially induce endogenous IL-10 activity in epithelial cells and (3) TNFα- and IL-1β-induced endogenous IL-10 differentially regulates the expression of PRRs and their negative regulators of signalling. Macrophage—epithelial cell co-cultures, however, demonstrated that: (1) M1/Caco-2 resulted in an inflammatory environment (TNFα, IL-6, IL-8), that effectively reduced barrier integrity (TEER and ZO-1), whereas exogenously added IL-10 augmented barrier integrity; (2) M2/Caco-2 generally maintained barrier integrity; (3) LcS rescues barrier integrity in M2/Caco-2 co-culture in the presence or absence of LPS, but not in M1/Caco-2; (4) LcS suppresses LPS-induced TLR2, TLR4 and MD-2 and differentially regulates CD14, NOD2, TLR9 and Tollip expression dependent on macrophage subset in co-culture; and finally, (5) LcS differentially modulates co-culture pro- and anti-inflammatory cytokines, dependent on macrophage subset and LPS stimulation.

The commensal bacterium, *L. rhamnosus* (LR), has been demonstrated to induce IEC IL-1β, TNFα, MCP-1 mRNA which fails to fully translate to secreted protein, where TNFα and IL-6 are not secreted, and MCP-1 is suppressed; only IL-1β is secreted. This relative unresponsiveness is also observed whereby LR suppresses *Bacteroides ovatus* induced IL-6 [38]. IECs are well established to exhibit a quiescent state with respect to cytokine production in a safe, homeostatic environment, whereas they rapidly respond to inflammatory cytokines such as TNFα and IL-1β, secreting IL-6 and IL-8. Considering that TNFα and IL-1β predominance has been described in IBD [39] and that these pro-inflammatory cytokines are readily expressed upon macrophage activation, it was important to define the inflammatory environment added to epithelial cells and the contribution of these epithelial cells to the inflammatory environment upon activation by LcS. LcS augmented both TNFα- and IL-1β-induction of IL-6, IL-8 and TNFα. Most importantly, LcS augmented TNFα-induced IL-10, whereas it suppressed IL-1β-induced IL-10 (Table 2). This observation alone is indicative that in the case of Crohn’s Disease (CD) patients, whose pathology is predominated by TNFα, LcS may exhibit a beneficial anti-inflammatory effect by inducing IL-10. In the case of CD predominated by IL-1β, however, LcS suppression of IL-10 may serve to further exacerbate inflammation of the gut mucosa. This only serves to reinforce the need to characterise the CD patient’s individual cytokine profile and that subtle differences in these cytokines can result in differential responsiveness to probiotic strains.

The crosstalk between epithelial cells and macrophages has been demonstrated to induce IL-10 in a TLR4-dependent manner, suggesting that in homeostatic conditions, both IL-10 and pathogen sensing were interlinked [40]. Based on the anti-inflammatory ability of IL-10 to suppress TNFα [37], the use of neutralising anti-IL-10 antibody, and a corresponding elevation in TNFα secretion, is an efficient method in detecting endogenous cell-associated activity of IL-10. This study clearly detected an endogenous IL-10 activity with unstimulated Caco-2 cells (Table 3). What was particularly remarkable was the observation that TNFα augmented endogenous IL-10 activity, whereas the IL-1β treatment failed to induce an endogenous anti-inflammatory IL-10 activity and even suggested a pro-inflammatory effect seen by decreased TNFα secretion upon neutralisation of IL-10. Such a response may go some way to explain the lack of success in using IL-10 in the treatment of CD, which was shown to induce pro-inflammatory IFNγ cytokine production [41]. As with the induction of secreted IL-10 above, the predominance of either TNFα or IL-1β in the inflammatory environment of CD patients, might effectively help or hinder inflammation. In this case, again IL-1β environments exacerbate inflammation through suppression of both secreted IL-10 and endogenous IL-10 activity.

This endogenous IL-10 activity also significantly affected the pathogen sensing ability of Caco-2 IECs in both homeostatic, unstimulated, or pro-inflammatory (TNFα- or IL-1β-stimulated) environments (Table 4). In a homeostatic, unstimulated environment, defined by PRR hyporesponsiveness [17,18], a measurable endogenous IL-10 activity suppressed bacterial recognition of LTA, LPS, PGN and unmethylated CpG DNA by TLR2, TLR4, CD14, MD2, NOD2 and TLR9, whereas it further reinforced a state of tolerance via inducing the negative regulator, Tollip. TNFα-induced endogenous IL-10 further suppressed TLR2, CD14, NOD2 and TLR9 but suppressed TLR4 expression to a level lower than unstimulated Caco-2 cells, whereas IL-1β-induced IL-10 suppressed TLR4, TLR2 and NOD2. This is indicative of IECs expressing a differential amplitude of bacterial sensing molecules, where TNFα-stimulated IECs express a weaker suppressive IL-10 activity on TLR4, whereas IL-1β-stimulated IECs express a strong suppressive IL-10 activity on TLR2, TLR4 and NOD2. In addition, both TNFα- and IL-1β-induced endogenous IL-10 negatively regulated Tollip (negative regulator of TLR signalling) expression. This will have a dramatic effect on the ability of IECs to recognise and respond to bacteria. In the presence of a TNFα inflammatory environment, the weaker IL-10 suppression of TLR4 is likely to bias recognition to Gram negative bacteria whereas the stronger IL-10 suppression induced by IL-1β is likely to result in lower-level recognition of LTA, LPS and PGN in both Gram-positive and -negative bacteria. Thus, the inflammatory environment and endogenous regulatory capacity of IECs has a dramatic effect of bacterial recognition, responsiveness and consequently can partially contribute to selective bacterial dysbiosis.

Expanding on these observations, which investigated IECs in monoculture, it was important to undertake similar studies in co-culture models that approach the cellular organisation of the intestinal mucosa. Such modelling has already been established in several studies investigating macrophage-epithelial cell interactions and their effects on barrier integrity and cytokine production in both homeostasis and disease [11,12,13]. Few, if any studies, have adequately described this cellular crosstalk and how it is modulated by probiotic strains in the context of barrier integrity and pathogen sensing. Basic innate immune cell and barrier epithelial cell interactions were investigated by developing M1/M2 macrophage-Caco-2 cell co-cultures. As expected, co-culture of Caco-2 cells with the homeostatic, regulatory M2 macrophage subset, resulted in low-level secretion of pro-inflammatory cytokines, whereas the secretion of TNFα, IL-6 and IL-8 was significantly higher in co-culture with the proinflammatory M1 subset (Figure 1). This is indicative that pathological environments, where M1-like subset predominates, will result in an appreciable inflammatory cytokine profile. CD is known to pre-dispose to such an M1-biassed mucosal environment [42,43]; effectively breaking tolerance and significantly activating an uncontrolled inflammatory response. This co-culture displayed a marked effect on barrier integrity, where M1 macrophages, and the cytokines produced in this environment (IL-1β, TNFα and IL-8), reduced both TEER and expression of the tight junction protein, ZO-1; observations previously noted for TNFα and IL-1β in Caco-2 epithelial barriers and experimental intestinal inflammation models [44,45]. ZO-1 protein was also markedly suppressed by the inflammatory molecules IL-1β and LPS, where the barrier organisation of IECs was obliterated in the M1/Caco-2 co-culture and only partially modified in M2/Caco-2 co-culture. It is likely that the local presence of M1 macrophages, their inflammatory cytokines and inflammatory stimuli such as LPS, have a detrimental effect on barrier integrity. This reduced barrier integrity, effectively creating a positive cycle of barrier destruction through macrophage activation and dysregulated pro-inflammatory cytokine production. Any early local production of IL-10 may have a rescuing effect on barrier integrity; however, co-culture modelling demonstrated low-level secretion, if any, in both co-cultures and what IL-10 activity that is present may be unable to countermand the pro-inflammatory effects of the elevated secretion of TNFα, IL-6 and IL-8 in M1-dominated environments.

When applying the probiotic bacteria, LcS, to these co-culture models, LcS clearly rescues barrier integrity (TEER & Z0-1) in M2/Caco-2 co-cultures in the presence or absence of enteropathogenic LPS. In the case of M1/Caco-2 co-culture, LcS partially rescues ZO-1 expression but fails to repair IEC barrier integrity, measured by TEER (Figure 2). This is indicative that LcS administration may be beneficial in homeostatic M2-specific mucosal environments but is likely to be ineffective in the treatment of M1-biassed pathology, such as seen in CD. Once the mucosal barrier is breached, the presence of M1, pro-inflammatory cytokines and PAMPs such as LPS, effectively perpetuate mucosal damage, which may be further contributed to upon the addition of probiotics such as LcS. Both co-cultures exhibited a suppressed LPS-induction of TLR2, TLR4 and MD2 in response to LcS and differential CD14, NOD2, TLR9 and Tollip modulation, where CD14 and NOD2 were suppressed in M2- and CD14 was unchanged whereas NOD2 was augmented in M1 co-cultures (Figure 3). In addition, TLR9 was augmented in M2- and suppressed in M1-cocultures; whereas the negative regulatory molecule, Tollip, was unchanged in M2- and rescued, but to a final insignificant low-level expression, compared to control IEC levels in M1-cocultures. This suggest that LcS reduces sensing of Gram negative LPS in both M1 and M2 co-culture and differentially controls how intracellular bacteria are detected and responded to, as well as regulated by Tollip; effectively reinforcing a selective tolerisation in M2/Caco-2 co-culture and attempting to limit pro-inflammatory damage in M1/Caco-2 but failing to induce tolerance as a consequence of augmentation of NOD2 mRNA expression and an insignificant rescue of M1-mediated suppression of Tollip.

Finally, when considering the ability of LcS to modulate macrophage-IEC co-culture secretion of inflammatory cytokines, LcS differentially modulated these factors dependent on macrophage subset and presence or absence of enteropathogenic LPS. In the absence of LPS stimulation, LcS suppressed M1/Caco-2 IL-6, IL-8 and IL-10 with no effect on TNFα; whereas augmented M2/Caco-2 IL-6 and IL-10. This is suggestive that, in the absence of pathogenic LPS stimulation, LcS exhibits a partially tolerising effect in the M1-co-culture, whereas tolerises the major inflammatory cytokines (TNFα and IL-8) whilst maintaining and reinforcing the secretion of the regulatory cytokines, IL-6 and IL-10 in the M2-co-culture. In the presence of enteropathogenic LPS, LcS suppressed M1/Caco-2 IL-8 and IL-10 and augmented TNFα and IL-6, whereas IL-8 was augmented, and IL-6 suppressed in M1 monoculture [8]. Conversely, LcS suppressed TNFα and IL-8 whereas augmented IL-6 and failed to modulate IL-10 secretion by M2/Caco-2 co-cultures (Figure 4) and in contrast, suppressed IL-6 and augmented TNFα in M2 monoculture [8]. Thus, LcS potentially fails to tolerise TNFα and IL-6-mediated inflammatory events and partially suppresses IL-8-mediated responses in M1 co-culture, whereas it suppresses TNFα and IL-8-mediated inflammatory responses in M2 co-culture. This differential regulation of two main orchestrators of inflammation (TNFα and IL-8) is suggestive of reflecting different mechanisms and amplitude of inflammation, as well as a relative difference in neutrophil recruitment to the site of active inflammation.

These observations both compared and contrasted with previous investigations, which clearly point to probiotic effects being both strain- and context-dependent. In vivo oral administration of *L. casei* CRL431 induced both CD206 and TLR2 in innate cells within organized payers patches [46]. Similarly, *L. casei* MYLO1 was shown to induce IL-10 and Tollip via activation of TLR2 and TLR9, resulting in a protective response to inflammatory liver damage induced by ethanol-TLR4-NFκB-mediated axis of inflammation [47]. This was further reinforced by an investigation focused on *Salmonella typhimurium* infection, where *L. casei* CRL431 ingestion increased TLR2, TLR4, TLR9, IL-10 expression and decreased TNFα production, hence modulating inflammation to a more protective response [48]. Finally, probiotic strains of *Weissella cibaria* were shown, not only to suppress pro-inflammatory cytokine production in both macrophages and epithelial cells, but to also suppress nitric oxide production [49]. Such a finding is suggestive of selective modulation of M1 activity and may even re-programme the macrophage subset to an M2-like phenotype. These observations describe selective effects of probiotics on immune activation and pathogen sensing, whereas, to date, few studies have investigated the effects of probiotics in both a homeostatic and inflammatory pathogenic context by utilizing distinct phenotypic macrophage subsets and their crosstalk to intestinal epithelial cells and how it impacts on overall barrier functionality.

In summary, IEC monoculture barrier integrity is sensitive to pro-inflammatory cytokines (IL-1β & TNFα) which also differentially modulate endogenous IL-10 activity capable of suppressing PRRs and inducing PRR negative regulators, such as Tollip. In IEC-macrophage co-culture, M1s induce a pro-inflammatory cytokine profile, whereas M2s induce a more tolerised profile, reflected in differential barrier integrity: severely compromised in M1/Caco-2 co-culture. LcS differentially suppressed PRR expression but failed to adequately rescue Tollip in M1 co-cultures. This differential suppression is also observed in barrier integrity where LcS failed to recover epithelial cell ZO-1 and TEER. As a consequence of this, LcS fails to suppress M1/Caco-2 TNFα, which is further augmented in the presence of LPS and accompanied by suppression of the anti-inflammatory, IL-10. In contrast, LcS suppressed M2/Caco-2 TNFα and IL-8 whilst augmenting IL-10; a response that is relatively unchanged by LPS. Thus, in conclusion, the relative balance between pro- and anti-inflammatory cytokines, positive PRR signalling and negative PRR signalling, has a significant impact on epithelial barrier integrity and perpetuation of chronic inflammation; determined by macrophage subset polarisation, LPS-PAMP stimulation and the ensuing cytokine profile. All of which are regulatable by LcS, which is context specific, which can be both detrimental and beneficial in the treatment of chronic IBD, such as CD. Only personalised diagnostic medicine (with respect to cytokine and PRR profile) can suggest beneficial adoption of probiotic strains in the management of CD.

## Figures and Tables

**Figure 1 microorganisms-10-02087-f001:**
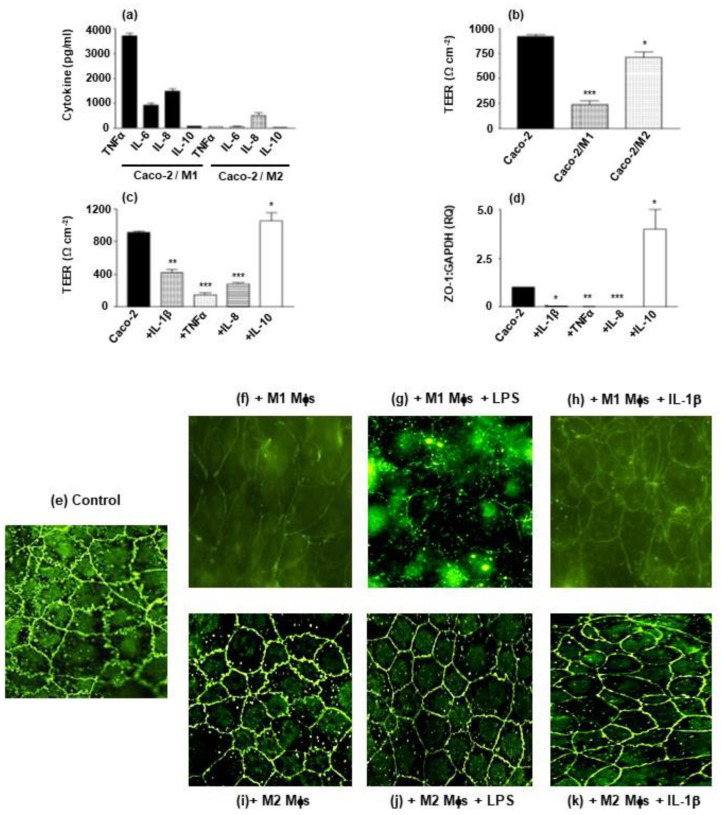
**Macrophage subset-epithelial cell co-culture and cytokine production determine barrier integrity.** Caco-2 epithelial cells were cultured to confluence (intact barrier) in transwell inserts and incubated in co-culture with M1 and M2 Mφ subsets (**a**,**b**) or Mφ-derived cytokines in the absence of Mφs (**c**,**d**). Co-cultures received LPS in the apical compartment and cytokine production was analysed in the basolateral compartment (**a**) and the corresponding TEER as a measure of barrier integrity (**b**). Effect of Mφ-derived TNFα, IL-1β, IL-8 and IL-10 on barrier integrity is presented as TEER in Ω cm^2^ (**c**) and expression of the tight junction molecule, ZO-1 mRNAexpression (**d**) is presented as relative expression (RQ, Arbitary Units) compared to GAPDH housekeeping gene expression. Mφ co-culture and effect of inflammatory stimuli is further represented by immunohistochemical characterisation of ZO-1 protein staining in the epithelial barriers incubated in co-cultures (**e**–**k**), where (**e**) Caco-2 epithelial barrier control, (**f**) Caco-2 + M1 Mφs, (**g**) Caco-2 + M1 + LPS, (**h**) Caco-2 + M1 + IL-1β, (**i**) Caco-2 + M2 Mφs, (**j**) Caco-2 + M2 + LPS and (**k**) Caco-2 + M2 + IL-1β. Data displayed is a representative experiment with triplicate samples for n = 4 replicate experiments (**a**–**d**) and n = 3 experiments (**e**–**k**). Significant effects of Caco-2/Mφ co-culture are compared to Caco-2 control (**b**) and cytokine effects compared to unstimulated Caco-2 control (**c**,**d**) and significance indicated as * *p* < 0.05, ** *p* < 0.01, *** *p* < 0.001.

**Figure 2 microorganisms-10-02087-f002:**
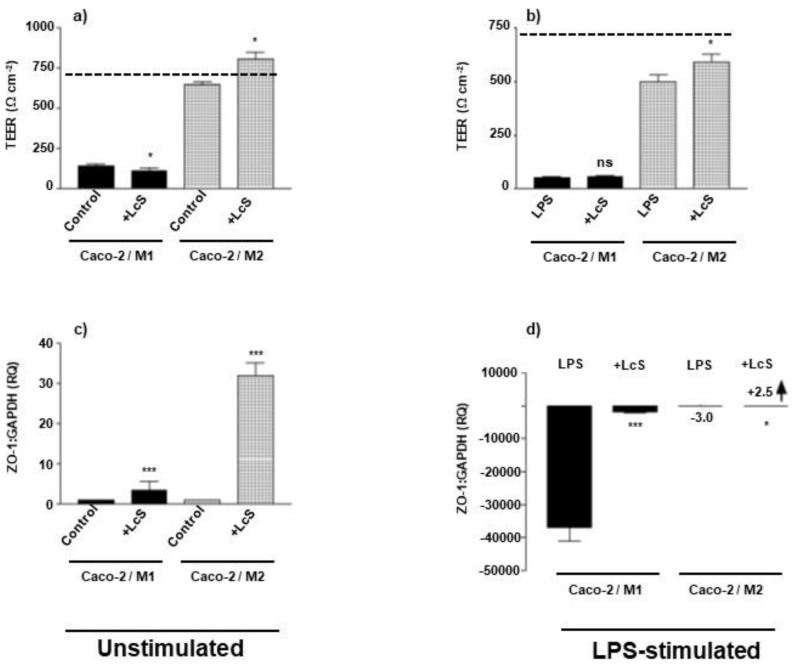
**LcS fails to rescue barrier integrity in M1 Macrophage subset-epithelial cell co-culture.** Caco-2 epithelial cells were cultured to confluence (intact barrier) in transwell inserts and incubated in co-culture with M1 (bold shading) and M2 (hatched shading) Mφ subsets in the presence (**b**,**d**) or absence (**a**,**c**) of apically applied LPS stimulation (100 ng/mL). Barrier integrity of co-cultures was analysed by measuring TEER (**a**,**b**) (with un-stimulated Caco-2 control barrier integrity TEER being indicated by dashed line) and presented in Ω cm^2^, and mRNA expression of the tight junction molecule, ZO-1 (**c**,**d**) is presented as relative expression (RQ, Arbitary Units) compared to GAPDH housekeeping gene expression. Data displayed is a representative experiment with triplicate samples for n = 3 replicate experiments. Significant effects of LcS treatment on Caco-2/Mφ co-cultures in the presence or absence of LPS stimulation are compared to non-LcS treated co-culture controls and significance indicated as * *p* < 0.05, *** *p* < 0.001 and ns, not significant and small-scale changes (**d**) are indicated by an arrow indicating augmentation/rescue and accompanied by the stated RQ relative expression values.

**Figure 3 microorganisms-10-02087-f003:**
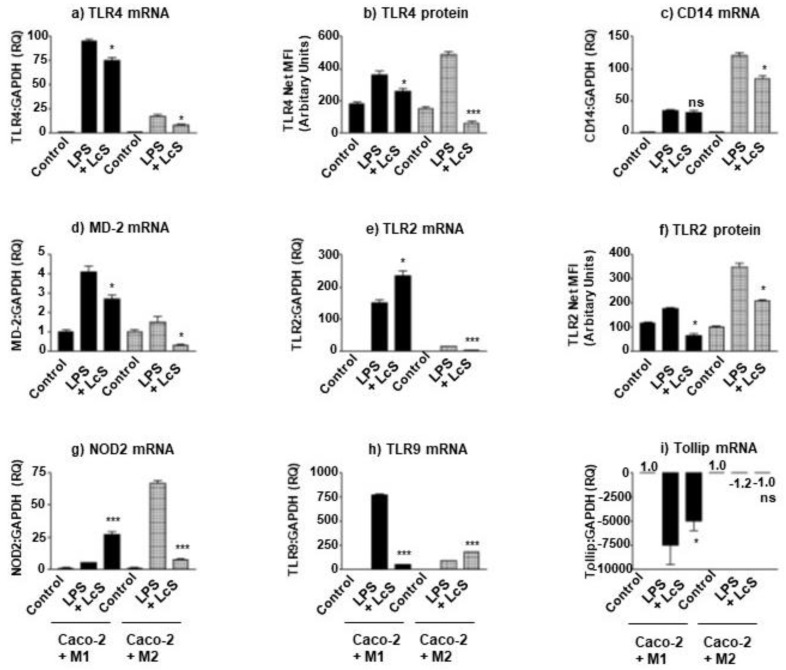
**LcS differentially modulates bacterial sensing molecules in M1 & M2 Macrophage subset-epithelial cell co-culture.** Caco-2 epithelial cells were cultured to confluence (intact barrier) in transwell inserts and incubated in co-culture with M1 (bold) and M2 (hatched) Mφ subsets in the presence of apically applied LPS stimulation (100 ng/mL) in the presence or absence of LcS. The LPS-receptor, TLR4, was measured as mRNA expression (**a**) and surface protein by flow cytometry (**b**). The mRNA expression of co-receptor molecules CD14 (**c**) and MD-2 (**d**) were also analysed, as was TLR2 mRNA (**e**), TLR2 surface protein (**f**), NOD2 mRNA (**g**), TLR9 mRNA (**h**) and the negative regulator of TLR signalling, Tollip mRNA (**i**). Gene expression is presented as relative expression (RQ, Arbitary Units) compared to GAPDH housekeeping gene expression and surface protein by flow cytometry as net MFI (Arbitary Units). Data displayed is a representative experiment with triplicate samples for n = 3 replicate experiments. Significant effects of LcS treatment on Caco-2/Mφ co-cultures in the presence of LPS stimulation are compared to non-LcS treated co-culture controls and significance indicated as * *p* < 0.05, *** *p* < 0.001 and ns, not significant and small-scale changes (**i**) are indicated by the stated RQ relative expression values.

**Figure 4 microorganisms-10-02087-f004:**
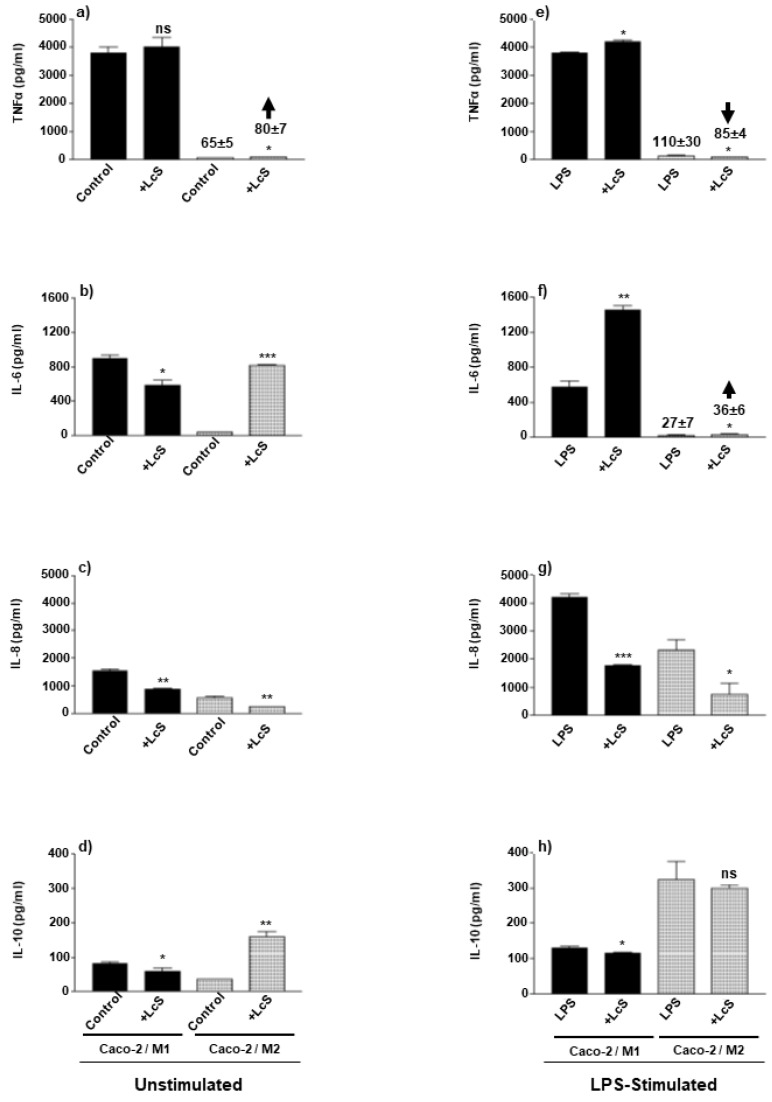
**LcS differentially modulates M1 & M2 co-culture cytokines: dependency on LPS stimulation.** Caco-2 epithelial cells were cultured to confluence (intact barrier) in transwell inserts and incubated in co-culture with M1 (bold) and M2 (hatched) Mφ subsets in the presence (**b**,**d**,**f**,**h**) or absence (**a**,**c**,**e**,**g**) of apically applied LPS stimulation (100 ng/mL) in the presence or absence of apically applied LcS. The pro-inflammatory cytokines, TNFα (**a**,**b**), IL-6 (**c**,**d**), IL-8 (**e**,**f**) and anti-inflammatory cytokine, IL-10 (**g**,**h**) were measured by sandwich ELISA and presented as protein production of the same secreted cytokines as the mean ± SD in pg/mL. Data displayed is a representative experiment with triplicate samples for n = 3 replicate experiments. Significant effects of LcS are compared to either the unstimulated control or LPS-stimulated control and are indicated as * *p* < 0.05, ** *p* < 0.01, *** *p* < 0.001 and ns, not significant and small scale changes (**a**,**e**,**f**) are indicated with arrow bars indicating augmentation or suppression and accompanied with an indication of cytokine concentration measured.

**Table 1 microorganisms-10-02087-t001:** Sequence of Real Time-PCR primers and estimated product size.

Target Gene	Forward Primer 5′-	Size (bp)	Reverse Primer 3′-	Size (bp)	Product Size (bp)
GAPDH	CTGCTCCTCCTGTTCGACAGT	21	CCGTTGACTCCGACCTTCAC	23	100
IL-6	TGGCTGCAGGACATGACAAC	20	TGAGGTGCCCATGCTACATTT	20	100
IL-8	TCAGAGACAGCAGAGCACACAA	22	GGCCAGCTTGGAAGTCATGT	20	100
IL-10	AGGAGGTGATGCCCCAAGCTGA	22	TCGATGACAGCGCCGTAGCCT	21	110
TNFα	ACATCCAACCTTCCCAAACG	20	GCCCCCAATTCTCTTTTTGAG	22	151
ZO-1	GCAATGGAGGAAACAGCTATATGG	24	TGAGGATTATCTCGTCCACCAGAT	24	104
TLR2	GGCATGTGCTGTGCTCTGTT	20	GGAGCCAGGCCCACATC	17	100
TLR4	AGCCCTTCACCCCGATTC	18	TAGAAATTCAGCTCCATGCATTG	23	100
TLR9	GGACCTCTGGTACTGCTTCCA	21	AAGCTCGTTGTACACCCAGTCT	22	151
CD14	ACCCTAGCGCTCCGAGATG	19	AGCTTGGCTGGCAGTCCTTT	20	100
MD-2	TGCACATTTTCTACATTCCAAGGA	24	ATAACTTCTTTGCGCTTTGGAAGA	24	100
NOD2	CAGAATTTCAAACGGCCTCACTA	23	ATGAAATGGAACTGCCTCTTGTG	23	102
Tollip	TCTCATGCCGTTCTGGAAAAT	21	TCACATCACAAAATGCCATGAA	22	110

Oligonucleotide sequences are presented for the forward and reverse primers for the cytokines (IL-6, IL-8, IL-10, TNFα), the PRRs (TLR2, TLR4, TLR9, NOD2), the PRR signalling molecules (CD14, MD2), tight junction molecule (ZO-1) and the negative regulatory molecule, Tollip, as well as the control housekeeping gene, GAPDH. Primer sequences were designed using Primer Express Software (Applied Biosystems, UK) for amplicon product size between 100–150 base pairs (bp).

**Table 2 microorganisms-10-02087-t002:** Heat-killed LcS differentially modulates epithelial cell cytokines induced by the pro-inflammatory cytokines, TNFα and IL-1β.

Readout/Treatment	Unstimulated	TNFα	TNFα + LcS	*p* Values
IL-8 mRNA IL-8 Protein	1.0 ± 0 15 ± 1	50 ± 1 25 ± 1	2700 ± 100 30 ± 2	<0.01 ** ns
IL-6 mRNA IL-6 Protein	1.0 ± 0 <7 ND	3 ± 1 <7 ND	4 ± 1 <7 ND	ns ns
TNFα mRNA TNFα Protein	1.0 ± 0 13 ± 1	500 ± 10 75 ± 3	2300 ± 200 60 ± 5	<0.001 *** <0.05 *
IL-10 mRNA IL-10 Protein	1.0 ± 0 20 ± 1	100 ± 1 175 ± 20	2100 ± 1 190 ± 10	<0.01 ** ns
**Readout/Treatment**	**Unstimulated**	**IL-1** **β**	**IL-1β + LcS**	***p* Values**
IL-8 mRNA IL-8 Protein	1.0 ± 0 15 ± 1	1000 ± 10 620 ± 20	2700 ± 100 780 ± 10	<0.01 ** <0.05 *
IL-6 mRNA IL-6 Protein	1.0 ± 0 <7 ND	3 ± 1 35 ± 2	4 ± 1 75 ± 4	<0.05 * <0.01 **
TNFα mRNA TNFα Protein	1.0 ± 0 15 ± 1	220 ± 10 55 ± 15	1100 ± 300 50 ± 3	<0.001 *** <0.05 *
IL-10 mRNA IL-10 Protein	1.0 ± 0 30 ± 1	10 ± 1 270 ± 10	2 ± 1 120 ± 20	<0.05 * <0.001 ***

Caco-2 epithelial cells were pre-treated with or without heat-killed bacteria, *Lacticaseibacillus casei* strain Shirota (LcS) for 18 h, followed by stimulation with either 10 ngmL^−1^ TNFα or 10 ngmL^−1^ IL-1β for a further 18 h. Induction of cytokine mRNA expression (IL-8, IL-6, TNFα and IL-10) is presented as relative expression (RQ, Arbitary Units), compared to GAPDH housekeeping gene expression and protein production of the same secreted cytokines as the mean ± SD in pg/mL. Data displayed is a representative experiment with triplicate samples for n = 3 replicate experiments. Significant effects of probiotic strains are compared to the cytokine-stimulated control (TNFα and IL-1β) and are indicated as * *p* < 0.05, ** *p* < 0.01, *** *p* < 0.001 and ns, not significant. ND: not detected; below lower limit of detection of ELISA(<7 pg/mL).

**Table 3 microorganisms-10-02087-t003:** Intestinal epithelial cells express an endogenous IL-10 activity dependent on stimulus.

Treatment	TNFα mRNA	TNFα Protein
Unstimulated +anti-IL-10	1.0 ± 0 6.1 ± 0.7 *p* < 0.001 ***	13 ± 1 315 ± 5 *p* < 0.001 ***
TNFα +anti-IL-10	4.0 ± 0.5 11.2 ± 0.4 *p* < 0.001 ***	80 ± 10 710 ± 20 *p* < 0.001 ***
IL-1β +anti-IL-10	5.0 ± 2.0 0.8 ± 0.1 *p* < 0.001 ***	330 ± 20 250 ± 15 *p* < 0.01 **

Caco-2 epithelial cells were pre-treated with or without 10 μgmL^−1^ anti-IL-10 neutralising antibody for 4 h, followed by stimulation with or without 10 ngmL^−1^ IL-1β or 10 ngmL^−1^ TNFα for a further 18 h. Induction of TNFα mRNA expression is presented as relative expression (RQ, Arbitary Units) compared to GAPDH housekeeping gene expression and TNFα protein secretion as the mean ± SD in pg/mL. Data displayed is a representative experiment with triplicate samples for n = 5 replicate experiments. Significant effects of endogenous IL-10 (anti-IL-10) are compared to the respective unstimulated or cytokine-stimulated control and significance indicated as ** *p* < 0.01, *** *p* < 0.001.

**Table 4 microorganisms-10-02087-t004:** Intestinal epithelial cell endogenous IL-10 activity regulates bacterial sensing molecules dependent on inflammatory stimulus.

Readout/Treatment	Control	+anti-IL-10	*p* Values
TLR4 mRNA—Unstim	1.0 ± 0	50 ± 1	<0.001 ***
+TNFα	5 ± 0.4	20 ± 0.5	<0.05 *
+IL-1β	20 ± 1	100 ± 1	<0.05 *
TLR2 mRNA—Unstim	1.0 ± 0	100 ± 5	<0.001 ***
+TNFα	5 ± 1	500 ± 10	<0.001 ***
+IL-1β	5 ± 1	900 ± 15	<0.001 ***
CD14 mRNA—Unstim	1.0 ± 0	70 ± 2	<0.001 ***
+TNFα	5 ± 1	120 ± 3	<0.001 ***
+IL-1β	70 ± 2	70 ± 1	ns
MD2 mRNA—Unstim	1.0 ± 0	3 ± 1	<0.05 *
+TNFα	0.8 ± 0.2	3 ± 1	<0.05 *
+IL-1β	4 ± 1.5	2 ± 1	<0.05 *
NOD2 mRNA—Unstim	1.0 ± 0	50 ± 6	<0.001 ***
+TNFα	15 ± 1	150 ± 5	<0.01 **
+IL-1β	70 ± 2	500 ± 10	<0.001 ***
TLR9 mRNA—Unstim	1.0 ± 0	90 ± 2	<0.001 ***
+TNFα	30 ± 2	110 ± 10	<0.001 ***
+IL-1β	3 ± 1	30 ± 4	<0.001 ***
Tollip mRNA—Unstim	1.0 ± 0	0.001 ± 0.0002	<0.001 ***
+TNFα	0.05 ± 0.002	0.10 ± 0.04	<0.05 *
+IL-1β	0.01 ± 0.001	0.10 ± 0.03	<0.05 *

Caco-2 epithelial cells were pre-treated with or without 10 μgmL^−1^ anti-IL-10 neutralising antibody for 4 h, followed by stimulation with or without 10 ngmL^−1^ IL-1β or 10 ngmL^−1^ TNFα for a further 18 h. Induction of bacterial sensing molecule mRNA expression (TLR4, TLR2, CD14, MD2, NOD2, TLR9 & Tollip) is presented as relative expression (RQ, Arbitary Units) compared to GAPDH housekeeping gene expression. Data displayed is a representative experiment with triplicate samples for n = 3 replicate experiments. Significant effects of endogenous IL-10 are compared to the respective unstimulated or cytokine-stimulated control and significance indicated as * *p* < 0.05, ** *p* < 0.01, *** *p* < 0.001 and ns, not significant.

## Data Availability

Not applicable.

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
