# Peer review of "Lacticaseibacillus casei Strain Shirota Modulates Macrophage-Intestinal Epithelial Cell Co-Culture Barrier Integrity, Bacterial Sensing and Inflammatory Cytokines"

_microorganisms, 2022, doi:10.3390/microorganisms10102087_

Round 1
Reviewer 1 Report
The authors investigate the influence of probiotic bacteria LcS on macrophages and epithelial cell function in vitro. This is an important study trying to unravel the interaction of (gut) bacteria with epithelial barrier function.
However, major comments have to be addressed before consideration of this manuscript for publication:
Interpretation of data from this study is rather difficult, because of the cell lines that were used. The authors draw their conclusions solely from experiments performed with differentiated M1 or M2-like THP-1 derived macrophages.
PMA-“differentiated” THP-1 cells where shown to express PRR e.g. TLR 4 only poorly and are generally low responders to PAMP stimulations. In fact, PMA differentiated THP-1 cells resemble a “weak” phenotype of M0 macrophages. In addition, PMA concentrations, length of time of PMA exposure, length of post-stimulation (recovery period) were shown to influence inflammatory responses and phenotype (PMID: 15153668, PMID: 26826276). Moreover, differentiation of THP-1 into M1 or M2 -like macrophages was shown to essentially be performed by stimulation with additional protocols (i.e. IFNgamma/LPS for M1 and IL-4/Il-13 for M2 like cells; PMID: 29520230). In contrast, the authors here use one concentration and one time-point (f.e. without a recovery phase) and only PMA or VitD3 and state a M1/M2-phenotypes of these cells.
Thus, basic characterization of (in this study) proposed M1 / M2-like phenotypes of THP-1 after PMA or VitD3 stimulation is missing in this manuscript. As the authors discuss phenotypic behaviour of macrophage-like THP1 cells, the starting cell populations have to be characterized in detail and these data (i.e. FACS) need to be shown at least in a supplemental data.
Moreover, additional experiments investigating LcS effects on differentiated primary human macrophages (M1 and M2, with LPS/IFNgamma and IL-4) would be important to provide proof of concept of modulation of macrophage functions.
Figures:
All Figures show a very low resolution and are hard to interpret.
For example, in Figure 3 resolution is too low to identify the axis labelling
Figure 4: Is the labelling in this figure correct?.
Fig.4f: in the legend text for (f) IL-8 measurement is described but in the graph the axis is marked “IL-6”?. Moreover, increase of cytokine level in 4f should be presented with “splitted” axis to see cytokine increase for M2 macrophage co-culture experiments.
For FACS experiments, gating strategies and representative FACS data (FACS blots, Histograms) should be presented at least in supplementary data.
Author Response
The authors wish to thank the reviewers for their diligence and real attention to detail in their feedback regarding this manuscript. We have tried to address each point raised, hopefully to your satisfaction. This has resulted in several areas in the mansuscript that have been either altered or added to. In addition, we have tried to address the issue of macrophage phenotyping by the addition of a comprehensive table of markers alongside both our published investigations and that of others, to demonstrate phenotypic segregation of subsets and similarity between PMA and IFNg/LPS, GM-CSF M1 macrophages and VD3 and IL-4/IL-13, M-CSF M2 macrophages; effectively demonstrating an extrapolation of THP-1 cell line derived macrophage subsets to that of M1/M2 derived from primary monocyte sources.
Reviewer 1
The authors investigate the influence of probiotic bacteria LcS on macrophages and epithelial cell function in vitro. This is an important study trying to unravel the interaction of (gut) bacteria with epithelial barrier function.
Reviewer Comment: However, major comments have to be addressed before consideration of this manuscript for publication: Interpretation of data from this study is rather difficult, because of the cell lines that were used. The authors draw their conclusions solely from experiments performed with differentiated M1 or M2-like THP-1 derived macrophages.
PMA-“differentiated” THP-1 cells where shown to express PRR e.g. TLR 4 only poorly and are generally low responders to PAMP stimulations. In fact, PMA differentiated THP-1 cells resemble a “weak” phenotype of M0 macrophages. In addition, PMA concentrations, length of time of PMA exposure, length of post-stimulation (recovery period) were shown to influence inflammatory responses and phenotype (PMID: 15153668,PMID: 26826276). Moreover, differentiation of THP-1 into M1 or M2 -like macrophages was shown to essentially be performed by stimulation with additional protocols (i.e. IFN gamma/LPS for M1and IL-4/Il-13 for M2 like cells; PMID: 29520230). In contrast, the authors here use one concentration and one time-point (f.e.without a recovery phase) and only PMA or VitD3 and state a M1/M2-phenotypes of these cells. Thus, basic characterization of (in this study) proposed M1 / M2-like phenotypes of THP-1 after PMA or VitD3 stimulation is missing in this manuscript. As the authors discuss phenotypic behaviour of macrophage-like THP1 cells, the starting cell populations have to be characterized in detail and these data (i.e. FACS) need to be shown at least in a supplemental data.
Author Response: The senior author has been working with differentiated THP-1s for over 20 years now. I agree with the reviewer that the differentiation protocol, as opposed to an activation protocol, needs to be fully characterised. Over the years, our group and many others have described both PMA and VD3 differentiated THP-1 macrophages. As the reviewer correctly indicates, a resting period is important for PMA, as this is a DAG analogue and can directly activate PKC, which is directly involved in the expression of several pro-inflammatory cytokines. As such, without a washout phase, we cannot be sure as to whether we are considering macrophage polarisation through longer-term differentiation or short-term activation. It has taken some time to refine the differentiation protocol to observe the phenotypes now indicated in the manuscript (methods section 2.2, paragraph 2, lines 141-145); thus optimising the protocol as: M1-like macrophages – 3-4 days PMA 25ng/ml, followed by a 24-48 hr PMA washout period prior to experimentation. M2-like macrophages – 7-8 days 1nM 1,25-dihydroxy Vitamin D3, with VD3 refresh at day 4 and washout prior to experimentation. The indicated phenotype: M1-like macrophages (iNOShi Arg- CD206- Phagocytosislo TLR4hi TNFahi IL-6hi IL-8hi IL-10lo); M2-like macrophages (iNOSlo/neg Arg+ CD206+ Phagocytosishi TLR4lo TNFalo IL-6lo IL-8lo IL-10hi). Such characterisation of both phenotype and methodological protocol has developed over several PhD projects and several published investigations. We have not published a protocol paper as would not significantly vary from other investigations from other laboratories. To attempt the full phenotyping for each study will impair both limited cell and financial resources required to undertake the focus of this investigation. These observations have partially been backed up by our studies and that of other groups for both IFNg/LPS (or GM-CSF) and IL-4/IL-13 (or M-CSF) and comparison to primary monocyte-derived macrophages using similar differentiation/polarisation factors. As a resource for phenotyping of THP-1-derived macrophages, we have included a phenotypic summary table and accompanying reference sources as a supplementary table/reference list at the end of the manuscript. One such attempt to draw our observations with that of others was published in the PhD thesis of Strachan, A. in my laboratory – I have used this information for the supplementary section and as such, have added his name to the author list.
Reviewer Comment: Moreover, additional experiments investigating LcS effects on differentiated primary human macrophages (M1 and M2, withLPS/IFNgamma and IL-4) would be important to provide proof of concept of modulation of macrophage functions.
Author Response: This manuscript is very much focussed on an in vitro model of the epithelial barrier-immune macrophage interaction and how it may be manipulated by probiotic products, with a view to regulating inflammatory or tumour environments in the gastrointestinal tract. The reviewers make an important point as to validity of using cell lines instead of primary cells. The use of primary cells is the ideal, but to undertake this, it will require access to a large volume of blood packs/buffy coats and the appropriate purification of PB-monocytes via centrifugal elutriation to prepare the required numbers of differentiated macrophages to undertake such a study. For this reason, the investigation was started by using a readily available and renewable source of cell line cells. If large financial investment was forthcoming, then paralleling this study with a primary cell source model would be the ideal: an approach that I may be able to justify with prior peer-reviewed publication using this cell line model.
Reviewer Comment: Figures: All Figures show a very low resolution and are hard to interpret.
For example, in Figure 3 resolution is too low to identify the axis labelling
Author Response: The authors apologise for putting the reviewers through this. All figures have been checked and reformatted to a higher specification/resolution.
Reviewer Comment: Figure 4: Is the labelling in this figure correct?.
Fig.4f: in the legend text for (f) IL-8 measurement is described but in the graph the axis is marked “IL-6”?. Moreover, increase of cytokine level in 4f should be presented with “splitted” axis to see cytokine increase for M2 macrophage co-culture experiments.
Author Response: Have clarified labelling on all figures and included numerical figures where values are too low for clear visibility when considering magnitude of response observed for other treatments in the same graph.
Reviewer Comment: For FACS experiments, gating strategies and representative FACS data (FACS blots, Histograms) should be presented at least in supplementary data.
Author Response: As indicated, we do not routinely carry out a full phenotyping experiment alongside the investigation detailed in this manuscript, due to cell number and financial constraints. We have established phenotypic data over a period of several years with the segregation of phenotypes indicated earlier and added in the methodology. When we have carried out flow cytometric phenotyping, we have detailed the staining of macrophage markers such as CD14, CD68, CD163, CD206, TLR2 and TLR4. In all cases, the flow cytometer was set up for live gating on the basis of FSC/SSC characteristics, to effectively exclude dead cells and cellular debris. Positive staining was then detected by collecting live-gated MFI data of the positive staining (anti-marker antibody) and presenting as net MFI upon subtraction of isotype-matched irrelevant marker antibody. This staining protocol has been established over several PhD projects supervised by the senior author and have been made available through publication of their respective thesis; some of which are referred to in the supplementary section focussed on macrophage phenotyping. As such, although we have tried to positively respond to all reviewers suggestions/comments, we do not think that it was appropriate here and have left the TLR data presented in figure 3 as MFI data in the format of a histogram.

Reviewer 2 Report
General comments
The manuscript aimed to evaluate the immunomodulatory effect of the probiotic strain, Lactobacillus casei Shirota (LcS), on epithelial cell – macrophage co-cultures with respect to mucosal barrier integrity, bacterial sensing and inflammatory cytokine production. The topic of the manuscript is very interesting and is also in line with the current interest on better understand how probiotics can modulate mucosal responses of the gut. In my opinion the manuscript needs some minor revisions to be considered suitable for publication.
Few suggestions/critical advice are reported below:
- LcS is a well-known probiotic strain largely use for its health-promoting properties in food products that usually deliver LcS live cells, but this study is focused on the effects of heat-killed cells because the authors prefer to use heat-killed bacteria to avoid “any effects of lactic acid produced by live LcS on cell viability, hence indirectly modulating cytokine responses..”. About that, in my opinion it could be interesting to evaluate or at least compare the effects of LcS live cells in modulating the mucosal response, thus I’m wondering if the authors have already tested live LcS cells and/or if they have previously found some negative effects on the use of live LcS cells in similar experiments that lead them to prefer heat-killed bacteria in this study.
KEYWORDS: keyword 1 is missing
TITLE/ABSTRACT/INTRODUCTION: Lactobacillus species nomenclature has been recently reclassified, Lactibacillus casei has renamed as Lacticaseibacillus (Lcb.) casei. Please update the nomenclature with the current one. You can check new microorganisms names at the following link http://lactotax.embl.de/wuyts/lactotax/
Line 53: As the first time that the name of Lacticaseibacillus casei occurs within the text, please mention it without abbreviation.
Line 71: Please replace the reference Haller et al. 2000 with a sequential number according to the reference list.
Line 80: As the first time that the name of Escherichia coli occurs within the text, please mention it without abbreviation.
Line 89 and line 92: To help the reader I suggest explaining molecules abbreviation the first time that they are mention within the text, such as peptidoglycan (PGN), lipotechoic acid (LTA) Toll inhibitory protein (Tollip).
Matherial and methods section: I suggest checking unit of measurement that should be written all in the same way
Line 125, 127: please check some typing errors
Table 1/Table 2/Table 3/Table 4: I suggest replacing the tables with higher resolution images or with Word tables
Figures and graphs: I suggest using images with higher resolution
DISCUSSION: Although the manuscript is well-written and results are well-reported, I would improve the discussion by reporting/comparing these results with more other studies investigating LcS and/or other probiotics strains (from line 589 to line 659 only one study by the same author is reported).
Author Response
The authors wish to thank the reviewers for their diligence and real attention to detail in their feedback regarding this manuscript. We have tried to address each point raised, hopefully to your satisfaction. This has resulted in several areas in the mansuscript that have been either altered or added to. In addition, we have tried to address the issue of macrophage phenotyping by the addition of a comprehensive table of markers alongside both our published investigations and that of others, to demonstrate phenotypic segregation of subsets and similarity between PMA and IFNg/LPS, GM-CSF M1 macrophages and VD3 and IL-4/IL-13, M-CSF M2 macrophages; effectively demonstrating an extrapolation of THP-1 cell line derived macrophage subsets to that of M1/M2 derived from primary monocyte sources.
Reviewer 2
The manuscript aimed to evaluate the immunomodulatory effect of the probiotic strain, Lactobacillus casei Shirota (LcS), on epithelial cell – macrophage co-cultures with respect to mucosal barrier integrity, bacterial sensing and inflammatory cytokine production. The topic of the manuscript is very interesting and isalso in line with the current interest on better understand how probiotics can modulate mucosal responses of the gut.
Reviewer Comment:
In my opinion the manuscript needs some minor revisions to be considered suitable for publication.
Few suggestions/critical advice are reported below:
- LcS is a well-known probiotic strain largely use for its health-promoting properties in food products that usually deliver LcS live cells, but this study is focused on the effects of heat-killed cells because the authors prefer to use heat-killed bacteria to avoid “any effects of lactic acid produced by live LcS on cell viability, hence indirectly modulating cytokine responses..”. About that,in my opinion it could be interesting to evaluate or at least compare the effects of LcS live cells in modulating the mucosal response, thus I’m wondering if the authors have already tested live LcS cells and/or if they have previously found some negative effects on the use of live LcS cells in similar experiments that lead them to prefer heat-killed bacteria in this study.
Author Response: The authors agree with this reviewer’s comment raised and appreciate that probiotics are defined as live microbes conferring a health benefit to the host. There are two aspects that must be considered: that most probiotic organisms die along transit and that more recently, it has been appreciated that both dead microbes and products derived from probiotics can confer health benefits and/or modulate the immune response. It is a simpler start to interpret probiotic effects on immune/barrier function if we neglect contributions of acidic metabolites and secreted products (SCFAs) derived from the utilisation of prebiotic substrates. Since this focus on HK-probiotics, we have since progressed our research to appreciate the effects of live probiotics and the products that they secrete. Preliminary results, quite surprisingly, indicate that production of lactic acid does not appreciably affect macrophage viability. There are both shared and contrasting immunomodulatory effects on macrophage functionality – this however, is to be the focus of another manuscript focussed on quite different strains of probiotic, so would not fit the structuring of this manuscript.
Reviewer Comment: KEYWORDS: keyword 1 is missing
Author Response: Sorry for the confusion. Keywords should be deleted after the original keywords subheading; so 5 keywords included 1 probiotics 2 macrophages 3 epithelial cells 4 cytokines 5 inflammation
Reviewer Comment: TITLE/ABSTRACT/INTRODUCTION: Lactobacillus species nomenclature has been recently reclassified, Lactibacillus casei has renamed as Lacticaseibacillus (Lcb.) casei. Please update the nomenclature with the current one. You can check newmicroorganisms names at the following linkhttp://lactotax.embl.de/wuyts/lactotax/
Author Response: The authors thank the reviewer for pointing this out and adding to the up-to-date nature of this manuscript. We were not aware of this change. This has been changed throughout the manuscript, to be presented as Lacticaseibacillus casei Shirota, when presented in full, but maintained as LcS, when abbreviated.
Reviewer Comment: Line 53: As the first time that the name of Lacticaseibacilluscasei occurs within the text, please mention it without abbreviation.
Author response: Line 54; microorganism written in full as Lacticaseibacillus casei strain Shirota (LcS), to accommodate presentation of new microbe nomenclature at first presentation.
Reviewer Comment: Line 71: Please replace the reference Haller et al. 2000 with a sequential number according to the reference list.
Author response: This reference has been added to reference list and labelled in text as reference 10. All following references have been re-numbered in in-text citation and reference section.
Reviewer Comment: Line 80: As the first time that the name of Escherichia coli occurs within the text, please mention it without abbreviation.
Author response: Escherichia presented in full, followed by abbreviated form in brackets.
Reviewer Comment: Line 89 and line 92: To help the reader I suggest explaining molecules abbreviation the first time that they are mention within the text, such as peptidoglycan (PGN), lipotechoic acid (LTA) Tollinhibitory protein (Tollip).
Author response: Have added full molecule names, followed by abbreviated forms in brackets, between lines 89 to 92.
Reviewer Comment: Material and methods section: I suggest checking unit of measurement that should be written all in the same way
Author response: at first, I could not see what was picked up. I then noticed the difference in presentation of units of time in hours or hrs, minutes or mins and cellsml-1 vs cells/ml. I have standardised to hrs, mins and cellsml-1 in methodology text. Checked the rest, could not find anything else. Thank you for your attention to detail. I have also standardised by making sure all units are presented after a space between numeral and unit.
Reviewer Comment: Line 125, 127: please check some typing errors
Author response: Corrected spelling of Dulbecco’s and adherent. Foetal and fetal are inter-changeable. I have stuck with foetal for consistency. Also corrected punctuation for Maria O’Connell, HNR, Cambridge. Thank you!
Reviewer Comment: Table 1/Table 2/Table 3/Table 4: I suggest replacing the tables with higher resolution images or with Word tables
Author response: have since changed tables to have greater clarity. I hope these are to your satisfaction. I acknowledge that the resolution of some of the figures is in need or sorting and will be responded to later.
Reviewer Comment: Figures and graphs: I suggest using images with higher resolution
Author response: Figures have been modified for clarity of labelling, which was relatively poor in first iteration of manuscript. For this, I can only apologise to the reviewer and hope that the current version is much better.
Reviewer Comment: DISCUSSION: Although the manuscript is well-written and results are well-reported, I would improve the discussion by reporting/comparing these results with more other studies investigating LcS and/or other probiotics strains (from line 589 to line 659 only one study by the same author is reported).
Author response: I agree with the reviewer that this section was rather focussed on our own findings and lacked an amount of context set by comparison to other published studies. In truth, there are very few, if any studies, which focus on the immune fate decisions via investigation of the cross-talk between different macrophage phenotypic subsets and intestinal epithelial cells. In an attempt to address this concern, I have included a further section at the end of this section indicated by the reviewer, which goes some way to show both similarities and differences observed for cytokine phenotypes and pathogen sensing in in vivo studies referring to the organisation of Payers patches and in vitro observations for macrophages and epithelial cells. This, I hope, will both facilitate rationale for our study and highlight both probiotic strain-dependence and possibly justify cell subset dependence in determining how mucosal tissue and co-culture models of such tissue respond in both a healthy/homeostatic or pathogenic/inflammatory context. As a result, this should reinforce our cautionary note on the use of probiotic strains in the management of inflammatory bowel disease: some strains will be beneficial, some will be detrimental.

Round 2
Reviewer 1 Report
I have no further comments.